# Potential Add-On Benefits of Dietary Intervention in the Treatment of Autosomal Dominant Polycystic Kidney Disease

**DOI:** 10.3390/nu16162582

**Published:** 2024-08-06

**Authors:** Erica Rosati, Giulia Condello, Chiara Tacente, Ilaria Mariani, Valeria Tommolini, Luca Calvaruso, Pierluigi Fulignati, Giuseppe Grandaliano, Francesco Pesce

**Affiliations:** 1Facoltà di Medicina e Chirurgia, Università Cattolica del Sacro Cuore, 00168 Rome, Italy; erica.rosati03@icatt.it (E.R.); giuliacondello.10@gmail.com (G.C.); chiara.tacente01@icatt.it (C.T.); ilaria.mariani04@icatt.it (I.M.); valeria.tommolini01@icatt.it (V.T.); luca.calvaruso@policlinicogemelli.it (L.C.); pierluigi.fulignati@policlinicogemelli.it (P.F.); giuseppe.grandaliano@unicatt.it (G.G.); 2Unità Operativa Complessa di Nefrologia, Dipartimento di Scienze Mediche e Chirurgiche, Fondazione Policlinico Universitario A. Gemelli IRCCS, 00168 Rome, Italy; 3Division of Renal Medicine, Ospedale Isola Tiberina—Gemelli Isola, 00186 Rome, Italy

**Keywords:** ADPKD, metabolic reprogramming, ketogenic dietary interventions, ketosis, beta-hydroxybutyrate

## Abstract

Autosomal dominant polycystic kidney disease (ADPKD) is the most common inherited cause of renal failure. The pathogenesis of the disease encompasses several pathways and metabolic alterations, including the hyperactivation of mTOR and suppression of AMPK signaling pathways, as well as mitochondrial dysfunction. This metabolic reprogramming makes epithelial cyst-lining cells highly dependent on glucose for energy and unable to oxidize fatty acids. Evidence suggests that high-carbohydrate diets may worsen the progression of ADPKD, providing the rationale for treating ADPKD patients with calorie restriction and, in particular, with ketogenic dietary interventions, already used for other purposes such as in overweight/obese patients or in the treatment of refractory epilepsy in children. Preclinical studies have demonstrated that calorie restriction may prevent and/or slow disease progression by inducing ketosis, particularly through increased beta-hydroxybutyrate (BHB) levels, which may modulate the metabolic signaling pathways altered in ADKPK. In these patients, although limited, ketogenic intervention studies have shown promising beneficial effects. However, larger and longer randomized controlled trials are needed to confirm their tolerability and safety in long-term maintenance and their additive role in the therapy of polycystic kidney disease.

## 1. Introduction

Autosomal dominant polycystic kidney disease (ADPKD) is the most common, potentially lethal, monogenic inherited progressive kidney disease (frequency 1:400–1:1000) [1]. ADPKD may be caused by pathogenic variants in either PKD1 or PKD2 genes, which occur, respectively, in 78% and 15% of patients [2] and encode, respectively, for Polycistin 1 and 2, located in the primary cilia. During the course of the disease, numerous cysts develop in both kidneys due to an excessive proliferation of tubule epithelial cells. This proliferation results in a notable increase in organ volume, accompanied by fibrosis and the gradual replacement of the normal renal parenchyma by cysts, ultimately leading to progressive and relatively slow renal failure.

The remaining 5% of ADPKD cases are attributed to rare mutations in other loci [3]. These less commonly implicated genes include HNF1B, a transcription factor that regulates the expression of genes associated with polycystic kidney diseases such as PKHD1 and PKD; IFT140 and GANAB involved in the protein folding process; and DNAJB11 [4,5,6,7]. Mutations in these “minor” genes are responsible for atypical forms of ADPKD, also known as the ADPKD-like phenotypes.

### 1.1. Metabolic Reprogramming in ADPKD

Metabolic reprogramming is now considered a prominent feature of ADPKD and could offer novel therapeutic targets [2,8]. Increased glycolysis, reduced fatty acid (FFA) β-oxidation, and altered mitochondrial function have been observed both in vitro and in vivo in animal models of ADPKD and in tissues from patients with ADPKD [9].

Rowe et al. found enhanced aerobic glycolysis and lactate accumulation in cells of murine model of PKD and in human-derived ADPKD kidneys [8]. Metabolic alterations depend on the dysregulation of the ERK pathway, resulting in increased intracellular cAMP levels and activation of the mTORC1-glycolytic cascade while simultaneously inhibiting the AMPK axis [8]. This results in increased glycolysis as the primary energy source for PKD1 mutant cells, even under normoxic conditions, resembling the Warburg’s effect observed in cancer [8,10]. The upregulation of glycolytic markers was confirmed in studies from subsets of human microarray data and cystic kidneys of Pkd1 mutant mice [11] while the TAME-PKD trial was the first evidence of human urine samples supporting the finding that increasing ADPKD severity is associated with a metabolic shift toward increased aerobic glycolysis [12]. Assuming that PKD cells develop metabolic adaptations compatible with an increase in glycolytic activity, it has been proposed that PKD1 null cells are extremely sensitive to low glucose, and that short-term inhibition of glucose metabolism leads to apoptosis and slowing of cystic disease [8]. To prove this hypothesis, the high rate of glycolysis has been targeted with glucose analog 2-deoxyglucose (2-DG), which mimics glucose deprivation, in orthologous mouse models of ADPKD [8,13,14] with effective cystic amelioration and reduced proliferation rates. A phase 1 clinical trial was designed with 2-DG in an ADPKD cohort [10]. Increased levels of ATP and lactate have also been observed in human ADPKD cell cultures and are reduced following treatment with 2DG [15]. Different studies described reduced FFA oxidations and oxidative phosphorylation [16,17,18] as relevant metabolic alterations in ADPKD. Menezes et al. reported reduced oxidative phosphorylation in Pkd1−/− cells that had fatty acids as their primary energy source, suggesting that FFA β-oxidation is reduced and typically accompanied by a compensatory increase in glycolysis [16]. It has been suggested that inducing a state of ketosis, which involves reducing glucose availability and increasing the degradation of fatty acids, could deprive cells of their main energy source. As a result, tissues are forced to use ketone bodies like acetone, acetoacetic acid, and beta-hydroxybutyrate (BHB) for energy. This leads to a nutrient deficiency that activates AMPK and inhibits mTOR, thereby effectively reducing cyst growth. Additionally, the impaired metabolism of FFA in these cells may result in the excessive intracytoplasmatic accumu-lation of lipid droplets, potentially leading to lipotoxicity and cell death [19]. Mitochondrial defects have been described in PKD and may be caused by reduced function of the polycystin complex, defective FFA oxidation, or other still unknown mechanisms [9]. In particular, mitochondrial fragmentation results in decreased mitochondrial respiration and increased production of reactive oxygen species (ROS) [20]. Mitochondrial abnormalities and enhanced ROS production in PKD activate ERK1–ERK2 signaling, which leads to increased glycolysis and increased cell proliferation [21]. Furthermore, a link between oxidative stress and endothelial dysfunction has been described in progression of kidney injury with ADPKD [22]. Inflammation has an important role in the progression of PKD. This is supported by the presence of pro-inflammatory markers in ADPKD urine and renal cyst fluid [23], accumulation of inflammatory cells, and the role of renal macrophages in PKD progression [24,25].

To date, Tolvaptan is the only drug approved globally for the treatment of ADPKD [26], with evidence that it can slow cyst growth and the progressive decline of kidney function. However, the therapeutic indication is for patients with rapid disease progression and its use is limited by liver toxicity [27] and aquaresis, which affects patients’ quality of life, resulting in a high rate of treatment discontinuation [28]. For these reasons, ADPKD patients in clinical practice frequently ask whether there is a way to slow their disease by modifying their diet or lifestyle. There is growing evidence that dietary interventions can be a modifier of ADPKD disease; thus, in recent years, the research is highly invested in non-pharmacological approaches (Figure 1).

### 1.2. Pathophysiologic Rational

The high glucose dependence of cyst-lining cells has induced interest in the dietary approach as a targeted therapy for cystic amelioration. As discussed earlier, 2-DG can be compared to caloric restriction without reducing food intake given its capability to reduce metabolic rate addiction to glucose [10]. Moreover, different studies have found that overweight and obesity, which are often associated with impaired glucose tolerance, correlate with a high rate of cyst growth and greater annual percent increase in total kidney volume (TKV) [29,30]. This may be caused by continuous caloric excess, which strongly activates the mTOR pathway and suppresses AMPK, leading to cyst growth. It has been suggested that slight to moderate caloric restriction could activate AMPK and suppress mTOR, slowing disease progression [31]. Specifically, ketogenic diets have been tested in ADPKD, hypothesizing ketogenesis might prevent or slow cyst growth.

The use of the ketogenic diet in humans was initially designed to reduce seizure onset in drug-resistant childhood epilepsy [32], and it is also recommended in treatment of many other clinical conditions such as weight loss, cancer, cardiovascular, metabolic (obesity and diabetes), and endocrine (polycystic ovary) alterations [33]. By definition, ketogenic dietary interventions (KDIs) are high-fat, low-carbohydrate (CHO), and moderate-protein diets which must induce ketosis [34]. Ketosis is the physiological adaptation to carbohydrate restriction, enabling the organism to use fat from both dietary intake and body reserves as an energy source. Therefore, to compensate for the reduced energy supply from carbohydrates, the dietary fat intake is increased in typical ketogenic diets [35]. During ketosis, adipose cells release fatty acids, which the liver converts into the ketone acetoacetate and BHB. The latter, considered the main ketone, not only serves as an energy source but also acts as a signaling molecule with significant cellular effects, including potent anti-inflammatory properties [36,37,38].

## 2. Preclinical Trials

The rationale for using a ketogenic approach (such as time-restricted feeding, intermittent fasting, and ketogenic diet) in animal models of PKD has been explored by several works [39,40,41,42] (Table 1).

The first experiment was conducted by Warner et al., who introduced a 10–40% food restriction (FR) in two different mouse ADPKD models, the Pkd1^RC/RC^ and the Pkd2 ^WS25/−^ mice. FR resulted in complete inhibition of cyst development after 6 months, and when FR was initiated after the establishment of cystic disease, it led to a reduction in cystic burden and beneficial effects on kidney function, as evidenced by decreases in BUN and Cystatin C levels. Additionally, FR was associated with decreased kidney inflammation, fibrosis, and injury, with no evidence of severe hypoglycemia. Finally, it was observed that early and 6-month FR positively regulated and activated the LKB1-AMPK pathway while significantly reducing the phosphorylation of S6, a key downstream substrate of the mTOR-S6K pathway [39].

Concurrently, another study [40] reported similar findings using a 23% reduction in food intake with no change in the qualitative composition of the diet in a human orthologous mouse model of ADPKD. The study demonstrated that even mild food restriction profoundly affects polycystic kidneys, resulting in reduced cyst growth, proliferation, fibrosis, and preservation of renal function. The analysis of kidneys by immunoblotting and immunofluorescence microscopy revealed reduced mTOR activation. In particular, the two major branches of mTORC1 signaling, S6 and 4EBP1, were both suppressed in cystic cells by reduced food intake, suggesting that dietary restriction may have broader effectiveness compared to pharmacological mTOR inhibition with rapalogs, which primarily affects the S6 branch [43]. However, unlike the previous study by Warner et al. [39], this study did not observe an increase in LKB1/AMPK signaling in PKD kidneys after food restriction [40]. This discrepancy may stem from differences in trial settings and mouse models used.

An elegant multi-arm preclinical trial by Torres et al. [41] demonstrated the efficacy of various dietary interventions inducing ketosis, including time-restricted feeding (TRF), ketogenic diets (KDs), and supplementation with BHB, in controlling disease progression in rat and feline models of PKD. In Han:SPRD rats, TRF induced ketosis, resulting in significant inhibition of cystogenesis, cyst expansion, proliferation, and fibrosis. Moreover, TRF-treated male rats exhibited reduced activation of the mTOR pathway substrate S6 and STAT3, a driver of cystic progression [44]. Similar results, along with increased levels of phospho-AMPK, were observed in juvenile Han:SPRD rats fed with high-fat, very-low-carbohydrate KD. However, while adult male rats on KD showed a 35% reduction in total polycystic kidney mass, serum creatinine levels and the total number of cysts per animal remained unaffected. Importantly, supplementation of a normal high-carbohydrate diet with BHB replicated the favorable effects of dietary restriction. BHB not only serves as an energy source for cells, but also exerts complex effects on numerous signaling pathways implicated in PKD, including mTOR and AMPK [45,46]. Additionally, acute fasting was found to significantly reduce cystic area and kidney mass by disrupting epithelial cyst-lining cells, leading to the loss of cyst fluid and accumulation of cytoplasmic oil droplets, thereby contributing to lipotoxicity and cellular death.

Hopp and colleagues [42] conducted a preclinical trial using an ADPKD1 orthologous mouse model (Pkd1RC/RC) subjected to daily caloric restriction (DCR), intermittent fasting (IMF), and TRF compared to ad libitum (AL) feeding. Animals on DCR showed a marked decrease in cystic disease, as evidenced by reduced kidney weight, cyst number and size, and renal fibrosis compared to those on AL and alternate feeding regimens. In contrast to prior publications, they did not observe a reduction in p-S6/S6 levels in animals on DCR versus AL [39,40]. However, kidneys of animals on DCR showed a significant increase in p-AMPK/AMPK compared to animals on AL and TRF and a trend toward increase compared to those on IMF. Despite the beneficial effects on cystic disease, animals on DCR exhibited growth retardation, raising concerns about potential long-term health complications in normal-weight, adult Pkd1 mice [42].

Preclinical studies are consistent in showing that calorie restriction prevents and/or slows the progression of the cystic disease, as it leads to a reduction in cyst formation and growth as well as renal fibrosis. Those beneficial effects may be due to the induction of a state of ketosis, specifically an increase in BHB levels, which could act by modulating the mTOR and AMPK pathways known to be altered in ADPKD.

## 3. Clinical Trials

Leveraging the evidence obtained from studies on animal models, the application of KD approaches was tested in ADPKD patients [42,47,48,49,50,51] (Table 2).

A preliminary pilot study by Testa et al. [47] explored the tolerability and safety of a 3-month modified Atkins diet (MAD) in three adult patients with rapidly progressive ADPKD. MAD, characterized by a breakdown of total energy intake with approximately 5% from carbohydrates, 30% from proteins, and 65% from lipids, was found to be well-tolerated and satisfying by all patients. Although some challenges were reported, particularly related to dining out and side effects associated with achieving a state of ketosis, overall satisfaction remained high. Despite successfully inducing ketosis and reducing glucose availability, all patients experienced an increase in total cholesterol levels during the follow-up period. The same authors are conducting a clinical trial to investigate the effect of a mainly plant-based MAD in total kidney volume variation [52].

The feasibility of DCR and IMF was tested in overweight or obese ADPKD patients with normal to moderately declined kidney function [42]. Clinically significant weight loss occurred with both DCR and IMF; however, weight loss was greater, and adherence and tolerability were better with DCR. In fact, patients randomized to IMF more likely experienced hunger, fatigue, cold intolerance, irritability, and insomnia. Triglycerides, total cholesterol, and low-density lipoprotein cholesterol were only reduced in the DCR group, and neither group demonstrated a change in eGFR. The annual percent change in height-adjusted total kidney volume (htTKV) was qualitatively low in both groups compared to historical data, thus suggesting that slowed kidney growth correlated with body weight and visceral adiposity loss, independent of dietary regimen. There were some limitations to the study such as the small sample size, the fact that htTKV data were a priori exploratory endpoint, and absence of a control group [42].

Strubl et al. [48] published a retrospective study based on self-reported observation and medical data of 131 ADPKD patients who followed KD and/or time-restricted diets (TRD) for at least 6 months. The primary endpoint was collecting patient-reported experiences about safety, feasibility, and possible beneficial effects of these dietary regimens. The results showed 80% of participants reporting an overall improvement in their quality of life. Almost all the participants significantly loss weight, with a weight reduction more pronounced in those following a KD compared to TRD. Similarly, the amelioration of disease related symptoms such as flank/back pain, fatigue, and abdominal fullness was better with KD compared to TRD. Concerning renal function, the paired analysis indicated stabilization of eGFR with small increases of 3.6 mL/min/1.73 m^2^ in the mean average, and individuals with documented ketosis had greater increases by 7.3 mL/min/1.73 m^2^, which positively correlated with the corresponding average serum BHB levels [48]. It is essential to evaluate changes in kidney function with greater accuracy by using creatinine clearance or kidney scintigraphy. This helps ascertain whether the observed improvement is real or stems from hyperfiltration or malnutrition, both of which are undesirable effects of a ketogenic regimen [19]. As side effects, participants in KD more frequently reported fatigue, hunger, and “keto flu” symptoms (headache, nausea, weakness, and blurred mind), which disappeared shortly after adaptation to ketosis. In addition, 22% of participants showed increased levels in total and LDL cholesterol levels, which were significantly higher in the KD cohort. In conclusion, KDIs appeared feasible for PKD patients in this trial. However, participants reported greater difficulty in long-term maintenance, particularly with the KD regimen. While these results are promising, it is worth noting the retrospective design of the study, the lack of a control group, and that all data rely solely on self-reported information from patients, thus rendering its reliability quite limited.

An interesting real-life experience trial examined the effects of a plant-focused ketogenic diet (PFKD) in ADPKD patients [49]. This dietary program, known as Ren.Nu, incorporates a medical food termed KetoCitra^®^, containing BHB and citrate, in formulation with potassium, calcium, and magnesium. Taken with meals, this medical food inhibits the absorption of dietary oxalate and inorganic phosphate while providing an alkaline load. The rationale behind the Ren.Nu program was to counteract metabolic abnormalities in ADPKD, including metabolic acidosis, excessive urine acidification, hypocitraturia, and an elevated risk of kidney stone formation, all while maintaining a daily protein intake of ≤0.8 gm/kg [49]. Additionally, plant-based diets are recognized for their renoprotective properties in chronic kidney disease [53]. Results from the study demonstrated improvement of PKD-related symptoms in 50% of participants, notably in flank pain and fatigue. The most frequent symptoms experienced during the initial two weeks of starting the PFKD were fatigue and mental fog, which resolved in nearly all patients once they transitioned into ketosis. Overall, 89% of participants reported weight loss, with significant decreases in fasting blood glucose levels observed. Additionally, all participants achieved and maintained a metabolic state of ketosis (defined by BHB ≥ 0.5 mM). Kidney function parameters improved as indicated by a mean creatinine decrease of 5.8% and a mean eGFR increase of 8.6% from the baseline average. In general, participants exhibited high levels of adherence and satisfaction, with 83% believing that this dietary program improved their overall health.

The first prospective interventional trial, RESET-PKD [50], investigated the short-term effects of ketogenic interventions, such as 3-day water fasting (WF) or a 14-day KD in 10 ADPKD patients. The study confirmed that a KD induces ketogenesis, but only WF resulted in a rapid and reversible decrease in total liver volume. This decrease was attributed to ketosis-induced depletion of liver glycogen stores. Neither intervention led to changes in total kidney volume (TKV), likely due to the short duration of the study [50]. Overall, 90% of patients reached the metabolic endpoint and/or the self-reported feasibility endpoint. Similarly to previous studies, total and LDL cholesterol increased significantly only in the ketogenic diet group and returned to baseline after switching back to a high-carbohydrate diet. Also, uric acid levels increased significantly in both groups, but gout attack or symptomatic kidney stone were not reported [50].

In a recent study, Cukoski et al. randomized ADPKD patients to receive ketogenic diet interventions, which included low-carbohydrate and high-fat KD or WF compared to routine dietary counseling [51]. Patients undergoing ketogenic diet interventions experienced weight loss, primarily driven by reductions in fat mass and body water content. However, significant weight loss was observed only in participants on the ketogenic diet compared to the control group. Additionally, in the ketogenic diet group, there was a significant decrease in mean ht-TKV, especially in individuals who reached higher biochemical thresholds of ketosis [54] compared to the control group. Instead, the significant reduction in liver volume upon KD was not influenced by the level of ketosis. Creatinine-based and cystatin C-based eGFR were increased in the KD group compared to the control group, but this difference was partially lost after the KD group switched back to a carbohydrate-rich ad libitum diet. However, according to the authors, it is not possible to discern whether the increase in eGFR resulting from KD indicates a beneficial effect on disease severity or simply reflects glomerular hyperfiltration. Additionally, participants in the KETO-ADPKD study exhibited relatively good kidney function on average, making it uncertain whether the same findings would apply to later stages of the disease. Further analysis of the lipid profile showed a significantly greater increase in total cholesterol, LDL cholesterol, non-HDL cholesterol, remnant cholesterol, VLDL, and ApoB levels in the KD group. Regarding safety, as expected, nearly half of the patients in the KD group experienced transient and mild symptoms of “ketogenic flu” at the induction of ketosis. In conclusion, the primary combined feasibility endpoint, which included both patient-reported feasibility and measured ketosis, was achieved only by the WF group. In the KD arm, only patient-reported feasibility was achieved, while the ketosis endpoint was not satisfied, partly due to an unintended requirement to reach high BHB cutoffs at three out of three on-diet visits [51].

Results from a recent randomized trial on the comparison between DCR and intermittent fasting in ADPKD patients are still unpublished [55], and new randomized controlled clinical trials are currently ongoing and are evaluating TRF [56], calorie restriction [57], and their effects on feasibility, tolerability, and slowing polycystic disease progression in overweight and obese ADPKD patients. A single interventional arm study on administration of oral ketone ester in ADPKD patients is not yet recruiting [58].

Although most of the available clinical trials are preliminary, with small sample sizes and short durations, they all agree on the feasibility and tolerability of the ketogenic regimen. However, concerns arise regarding its long-term maintenance. Weight loss and reductions in blood glucose levels are common findings across all studies, but it is unclear whether there are benefits on renal function and the reduction in kidney cyst growth. Some authors report a reduction in htTKV [42,51], while others observe no variation [50]. The increase in eGFR has been reported by some studies [48,49,51], but it was not possible to determine whether it was due to a real beneficial effect of the diet or if it was the result of hyperfiltration or malnutrition. Regarding height-adjusted total liver volume (htTLV), significant but reversible reductions have been observed, attributed to decreased hepatic glycogen reserves [50]. Some studies have reported a worsening of the lipid profile, with increases in total cholesterol and LDL cholesterol [41,47,50]. Regarding the safety of the ketogenic regimen, new symptoms classified as “ketoflu” have been reported, which are temporary and disappear within the first weeks of treatment once the state of ketosis is established [42,48,49,51], without reducing the overall satisfaction of patients with ketogenic approaches.

## 4. Side Effects of Ketogenic Diet

### 4.1. Metabolic Acidosis

An important concern associated with ketogenic diets is the promotion of acidosis relating to high protein intake, reduced dietary alkali intake derived from low or potentially low consumption of fruits and vegetables [28], and a weakly acidifying effect of ketones. In healthy subjects, ketogenic diets have no significant impact on serum bicarbonate levels [59,60,61], but it is unclear whether ketogenic interventions could significantly worsen metabolic acidosis or affect net acid excretion in chronic kidney disease and notably in patients with ADPKD. In fact, in these subjects, metabolic acidosis could hast the decline of renal function [62] and the development or worsening of renal bone disease, muscle wasting, hypoalbuminemia, muscle inflammation, protein malnutrition, and even mortality [63].

### 4.2. Kidney Stones and Hyperuricemia

A meta-analysis estimated that ≈8% of individuals on ketogenic diets develop kidney stones during a follow-up of ≈4 years [64]. Uric acid stones are the most frequently reported by individuals on a ketogenic diet, followed by calcium oxalate stones or mixed stones with calcium and uric acid [64], suggesting a potential link to animal protein intake as a known cause of high uric acid burden. The initial and transient rise in uric acid is concomitant with the rise in ketones, and it was postulated that the reason for this may be competition between uric acid and BHB for the same organic acid transporters, which are required for renal excretion [65]. This may represent a limitation since kidney stones are significantly more common in patients with ADPKD compared with the general population [66] with a higher frequency of uric acid stones [67,68]. Bruen at al. found a significant increase in uric acid resulting in a hyperuricemia in both groups after the KDIs, but no participant was found to have a new kidney stone after the start of ketogenic diet after the start of the ketogenic diet [49]. Other risk factors for nephrolithiasis are metabolic acidosis that occurs with ketogenic diets, which potentially reduce urine citrate, a well-known inhibitor of stone formation [69], and the reduced water consumption that occurs with the avoidance of fruits and vegetables, which are high in not only carbohydrates but also water [70].

### 4.3. Elevation of Low-Density Lipoprotein Cholesterol (LDL-C)

Several cited studies have observed reductions in significant cardiovascular risk factors, such as HDL cholesterol levels, triglycerides, BMI, waist circumference, and blood pressure. However, LDL-C is the only lipid marker consistently reported to be elevated in certain individuals following ketogenic diets, and it is associated with cardiovascular risk and atherosclerotic plaque growth [71]. In any case, transient increases in cholesterol/LDL are a well-reported and necessary effect of KDIs, indicating the successful depletion of adipose lipid stores. They have been shown to normalize again over time [72,73]. Additionally, most studies have only reported the standard LDL-C test, which does not differentiate between the harmful and more atherogenic small dense LDL particles and the benign, large, LDL particles [74]. A recent meta-analysis found that low-carbohydrate diets decrease harmful small dense LDL particles while increasing large LDL particles, suggesting a net beneficial effect on CVD risk [74]. However, elevated LDL-C levels are a clearly defined cardiovascular risk factor in clinical practice, regardless of their subtyping, and chronic kidney disease is a state of increased cardiovascular risk in general [75]. Considering the short duration of the studies, the long-lasting effect of the ketogenic regimen on dyslipidemia and CVD risk in ADPKD patients requires more long-term prospective trials. Additionally, it was reported that ketogenic diets can lead to prolonged QT time with an increased risk of cardiac arrhythmias [76].

### 4.4. “Keto Flu” Symptoms

"Keto flu” describes a variety of “flu-like” symptoms that occur in individuals who are entering a ketosis state for the first time. These symptoms, which are generally temporary, resolve within a few days and may include fatigue, lethargy, dizziness when standing, unusual dreams, muscle spasms/cramps, headache, brain fog, gastrointestinal symptoms, and/or nausea [35]. As with any dietary change, gastrointestinal symptoms such as diarrhea or constipation can be due to adaptations of the intestinal microbiome, e.g., in response to changes in fiber or fat intake. Other symptoms such as headaches and cramps are due to electrolyte changes due to depletion of glycogen stores that leads to excretion of associated water [35]. Despite potential beneficial effects, some studies revealed that individuals have poor adherence to maintaining the low-carbohydrate requirement of the ketogenic diet for long periods of time [77,78] and that in regard to weight loss, partly the diet may be due to nonfat loss, including muscle mass and body water losses [79].

## 5. Ketosis-Inducing Therapies: SGLT2 Inhibitors

The introduction of sodium-glucose transporter 2 inhibitors (SGLT2i) into clinical practice has significantly influenced the management of CKD and was rapidly integrated into clinical guidelines [80]. Gliflozins effectively mitigate declines in the eGFR, delay microalbuminuria, and limit proteinuria progression, favoring both diabetic and nondiabetic patients. The renoprotective effect of SGLT2is has been postulated to be partially mediated by the modest medication-induced ketosis [81,82,83]. This low-grade ketosis induced by SGLT2is may directly or indirectly benefit the kidney by serving as an energy source during stress and kidney injury, and through its anti-inflammatory, antifibrotic, and antioxidant effects [81,82]. Results from PKD animal models on slowing disease progression and cyst growth are controversial [84,85,86,87]. Unfortunately, ADPKD was an exclusion criterion in chronic kidney disease (CKD) trials [88,89]. Thus, efficacy and safety of SGLT2is remain unexplored in this population.

## 6. Dietary Counseling in ADPKD

### 6.1. Protein Intake

In ADPKD, a high protein intake induces hyperfiltration and increases vasopressin release, potentially leading to cyst growth and kidney function decline [90]. Aukema et al. showed in polycystic mice that a low protein regimen reduced cyst growth and kidney volume compared to a normal protein diet [91]. Studies in ADPKD patients show inconsistent results. According to some authors, it is not possible to demonstrate that a low protein intake is correlated with slower progression of CKD [92,93,94]. Conversely, a high urine-to-plasma urea ratio, which correlates with higher protein intake, has shown a strong association with the reduction in eGFR [95]. Even in the absence of clear evidence on protein restriction in ADPKD, it seems prudent to suggest a moderate protein intake of around 0.8 g/kg/day, as recommended for CKD patients [80].

### 6.2. Salt Intake

In the general population, excessive salt intake is correlated with increased cardiovascular diseases and mortality [96,97], while reducing salt intake has been shown to result in decreased systolic blood pressure [98], proteinuria [99], renal and cardiovascular events [100,101]. In patients with ADPKD, arterial hypertension contributes considerably to the increased cardiovascular morbidity, mortality, and disease progression [102]. It has been shown that blood pressure reduction leads to a 14% decrease in cyst growth [103] and the reduction in dietary sodium intake can significantly decrease vasopressin secretion in ADPKD patients [104,105]. In fact, salt intake seems to be associated with declining renal function in ADPKD through a mechanism involving increased vasopressin release [93]. Therefore, reducing salt may enhance the favorable and protective effect of renin–angiotensin–aldosterone system (RAAS) blockade and reduce harmful levels of vasopressin. In the CRISP cohort [94], higher urinary sodium excretion, used as a marker for dietary sodium intake, was correlated with the rate of TKV increase in relatively early stages of the disease [94]. Strong evidence came from a post hoc analysis of the HALT PKD trial [106], where the authors demonstrated a causal relationship between dietary sodium and kidney growth. They also suggested that lower salt intake (≤2.4 g/day) may be associated with a slower rate of eGFR decline, although this was only a trend and not a definitive finding [107]. These results suggest that dietary sodium restriction may be crucial in the therapeutic management of patients with ADPKD in terms of slowing disease progression and enhancing commonly used medications in ADPKD. To date, sodium intake recommendations for ADPKD patients rely on those provided by the guidelines for CKD patients [80,108,109], and should probably be <2–2.3 g of sodium per day.

### 6.3. Water Intake

In ADPKD, the urine concentrating deficit leads to compensatory increased vasopressin levels [110] which are detrimental [111], since vasopressin V2 receptor activation induces the production of both adenylyl cyclase and cyclic adenosine monophosphate [112,113]. Therefore, increasing water intake or decreasing osmolar intake may lower vasopressin concentration and slow disease progression [114], as demonstrated in the ADPKD animal model [115,116,117,118]. Several short-term human studies have shown that a low osmolar diet and increased water intake induced a decrease in urinary osmolality [119,120] and a decrease in copeptin, a surrogate marker of vasopressin [104,121]. However, long-term high water intake did not show a significant difference in copeptin levels compared to ad libitum water intake [122]. Higashihara et al. [123] investigated the effect of increased water intake on ADPKD disease progression in a 1-year non-randomized study and found enhanced disease progression, indicated by trends towards faster eGFR decline and TKV growth. Nevertheless, there are ongoing clinical trials to demonstrate the efficacy of high water intake in patients with ADPKD (ANZCTR12614001216606, NCT02933268, NCT03102632) [124,125,126]. The recommended fluid intake in ADPKD patients is still not exactly known. KDIGO guidelines recommend achieving a urine osmolality of ≤270 mOsmol/kg [28].

### 6.4. Caffein Intake

Caffeine stimulates the release of intracellular calcium and inhibits the activity of phosphodiesterases that hydrolyze cAMP, resulting in an increase in this second messenger [127], which appears to be involved in the growth of renal cysts in patients with ADPKD [128,129]. However, chronic administration of caffeine in murine models of ADPKD has not shown significant changes in GFR and cyst size [130]. Additionally, studies on ADPKD patients [131,132] have not demonstrated clinically relevant effects of caffeine intake on TKV or eGFR. Due to a lack of evidence, strictly limiting caffeine intake in patients with ADPKD does not seem necessary. Conversely, mild to moderate coffee consumption has been associated with metabolic and cardiovascular benefits [133]. Therefore, patients with ADPKD may benefit from moderate consumption of caffeinated beverages (200–250 mg per day).

## 7. Conclusions

Metabolic alterations seem to play a key role in the pathogenesis of ADPKD, and dietary manipulations targeting metabolic sensors may have a role in the prevention or treatment of this disease. The increasing interest in KDIs is based on the inhibiting effect of calorie restriction on the mTOR pathway [19]. The animal studies cited in this review showed that ketogenic interventions such as time-restricted feeding, administration of a ketogenic diet, or supplementation with exogenous BHB prevented kidney cyst disease progression by inhibiting cell proliferation, fibrosis, and cyst growth. Furthermore, in these animals, mTOR activity was inhibited, strengthening the hypothesis that the inhibition of this metabolic target may prevent or slow disease progression. Interestingly, Torres et al. [41] suggested that limiting the reduction in glucose availability may contribute to the efficacy of ketosis, but it is not as essential as the increase in BHB levels [41], induced to varying degrees by all ketogenic interventions. Remarkably, treatment with BHB alone, even in the context of a normal high-carbohydrate rodent diet, proved to be highly effective, suggesting that BHB itself is a major factor in suppressing PKD progression [41], probably because it is involved in improving the mitochondrial efficiency [134,135] notably altered in ADPKD and influences the mTOR and AMPK pathways [37,38,46]. Furthermore, BHB has potent anti-inflammatory properties, and ketogenic diets have been shown to reduce low-grade inflammation in obesity and other conditions [136].

Ketogenic dietary interventions in ADPKD subjects have demonstrated promising results in terms of general tolerability and feasibility. Key benefits include reductions in body weight, blood glucose levels, and improved blood pressure control. However, their effects on eGFR and the annual percent change in htTKV remain unclear. Comparable outcomes have been observed in studies involving patients with CKD attributed to other etiologies. KDs have been shown to significantly lower blood pressure, in some reports even better than the DASH diet [137]. Furthermore, ketogenic diets promote a natriuretic and diuretic effect similar to that demonstrated during starvation, partly caused by the typical rapid weight loss that occurs during the initiation of a KD [138,139]. Those effects may help alleviate sodium retention and improve systemic and glomerular blood pressure and may be particularly beneficial, since arterial hypertension can accelerate PKD disease progression. Different studies both in overweight/obese or and/or T2D patients have shown that the beneficial effect of low-carbohydrate diet was greater in individuals with lower starting baseline eGFR [140,141,142], and that there exists a dose-dependent association between ketosis and the increase in total eGFR slope in subjects followed up for 2 years [143]. So, it is reasonable to assume that low carbohydrate diet may be safe and effective even in patients with ADPKD and reduced eGFR, although the studies conducted so far in humans are of short term and report the mean eGFR > 50 mL/min/1.73 m^2^.

On the other hand, KD is not advisable to individuals with impaired kidney function [144], in part due to concerns about increased daily protein intake, ranging from 0.6 g/kg to 1.4 g/kg [145,146,147]. High protein intake has been associated with hyperfiltration, increased acid excretion, and potentially, a decline in kidney function [148], but a very-low-protein diet (0.28 g/kg/day) was associated with increased risk of death at a median follow-up of 3.2 years in nondiabetic CKD patients [149]. The only requirement for a ketogenic diet is that carbohydrate intake is restricted sufficiently and that triglyceride intake is correspondingly increased, irrespective of protein intake. Thus, ketogenic regimens could be implemented in combination with low-protein plant-dominant (PLADO) [150] and plant-focused diets (PLAFOND) [53], with the aim of reducing the daily protein load and preventing further acidosis and kidney stone formation. Plant-focused diets are already recognized as being beneficial in CKD [151], with appropriate adjustments to mitigate the risk of hyperkalemia. Results from the Ren.Nu Program [49] suggest the possibility of achieving the benefits of ketones by providing an external source of BHB, regardless of the diet composition, as already suggested in animal models [41], thereby overcoming the challenges of long-term adherence to ketogenic diets.

As suggested by Weimbs et al. [35], it would be appropriate to introduce the concept of ketogenic metabolic therapy rather than ketogenic diet. This therapy aims to switch the body’s metabolism to a state of ketosis without excessive food restrictions and micronutrient deficiencies and can be administered for long periods of time. The current nutritional recommendations for patients with ADPKD are largely based on that developed for CKD due to other etiologies. However, there are significant differences due to the unique characteristics of the disease; consequently, moderation in caloric intake has sensibly been recommended for the management of ADPKD [152]. Although the results on the benefits of the KD are promising and encouraging, there is still no recommendation to adhere to ketogenic regimens for the treatment of polycystic disease [28], and it is important to remember that KDs are not necessarily free of side effects and should be supervised by a physician. To date, there has been no evidence supporting strong recommendations for any dietary advice or interventions that can significantly and specifically modify the progression of ADPKD (Table 3).

Given the potential benefits of KDIs, larger and longer-term randomized controlled trials are needed for patients with ADPKD. These trials should examine renal hard endpoints such as rapid eGFR decline, annual htTKV change, initiation of dialysis or kidney transplantation, net acid excretion, kidney stone formation, and sodium retention. Additionally, the impact of ketogenic diets on metabolic reprogramming should be explored further. A comparison of the safety and efficacy of ketogenic nutritional therapy versus pharmacologic intervention that induce the raising of ketone levels, such as SGLT2i (not yet approved in ADPKD patients), could be of high interest. Metformin, a well-known oral hypoglycemic agent widely used for treating type 2 diabetes, has shown encouraging results in reducing cystogenesis in preclinical models of polycystic disease [153,154]. Metformin’s mechanism of action involves activating AMPK, which leads to decreased cell proliferation and fluid secretion, thereby reducing cyst growth [155,156]. Since both metformin and ketogenic dietary interventions (KDIs) activate the AMPK pathway, it is plausible they could have a combined effect in reducing cysts. However, published data do not show clear evidence of benefits on changes in ht-TKV or eGFR [157,158]. Further studies are ongoing (NCT04939935; NCT03764605). Finally, it is crucial to understand the additive role of concurrent KDIs and Tolvaptan, the only currently available treatment drug for patients with ADPKD. We currently lack solid evidence of the beneficial effect of combining a ketogenic diet with Tolvaptan. Nevertheless, it may be reasonable to hypothesize that they might have an additive effect, possibly acting on different molecular mechanisms to slow cyst growth.

## Figures and Tables

**Figure 1 nutrients-16-02582-f001:**
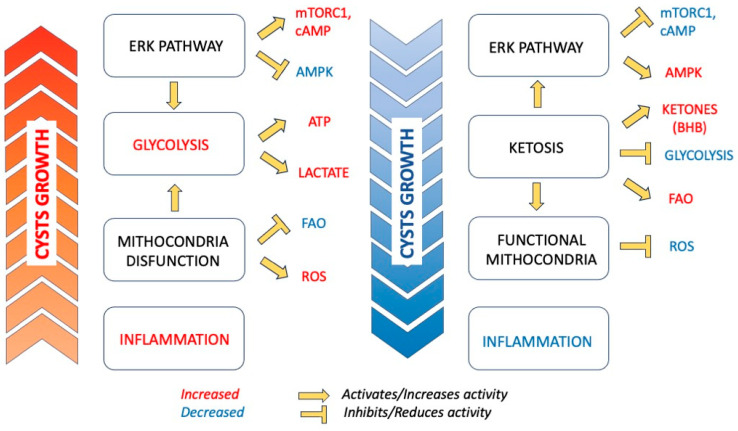
Metabolic Reprogramming in ADPKD and the Impact of Ketosis. Metabolic alterations in ADPKD drive cystogenesis and cell proliferation. Dysregulation of the ERK pathway activates the mTOR cascade and increases intracellular levels of cAMP while inhibiting the AMPK pathway. These changes lead to increased glycolysis, higher ATP levels, and extracellular lactate accumulation. Altered mitochondrial function results in elevated production of ROS and reduced fatty acid β-oxidation, further promoting glycolysis. Additionally, systemic inflammation in ADPKD sustains disease progression. Inducing a state of ketosis may help restore metabolic balance and slow cyst formation and growth. During ketosis, reduced glucose availability forces cells to utilize ketone bodies, particularly BHB, and fatty acids as energy sources. This shift restores mitochondrial function and reduces ROS production. Furthermore, reduced glycolysis leads to proper activation of the ERK pathway, resulting in decreased mTOR activation and increased AMPK activity. ADPKD, Autosomal Dominant Polycystic Kidney Disease; mTORC1, mammalian Target of Rapamycin Complex 1; cAMP, cyclic Adenosine Monophosphate; AMPK, Adenosine Monophosphate-Activated Protein Kinase; FAO, fatty acid β-oxidation; BHB, beta-hydroxybutyrate; ROS, Reactive Oxygen Species.

**Table 1 nutrients-16-02582-t001:** Preclinical trials investigating Ketogenic Dietary Interventions in Polycystic animal models. DCR, Daily Caloric Restriction; IMF, Intermittent Fasting; TRF, Time-Restricted Feeding; BHB, Beta-hydroxybutyrate; LKB1-AMPK, Liver Kinase B1—AMP Activated Protein Kinase; mTOR-S6K, Mechanistic Target of Rapamycin- S6 Kinase; p, Phosphorylated; STAT3, Signal Transducer and Activator of Transcription 3.

Authors, Year of Publication	Intervention	Animal Model	Duration	Findings
Warner et al., 2016 [39]	10–40% DCR versus ad libitum	Pkd1^RC/RC^ mice; Pkd2. ^WS25/−^ mice	From 6 wk to 7.5 mo	Reduced cyst area, kidney fibrosis, inflammation, and injury; Improved kidney function; Increased activation of LKB1-AMPK and reduced activation of mTOR-S6K.
Kipp et al., 2016 [40]	23% DCR versus ad libitum	Pkd1^cond/cond^:Nes^cre^ mice	Postnatal wk 5–12	Reduced cyst growth, proliferation and fibrosis; Maintained renal function; Reduced mTOR activation, not increased activation of LKB1-AMPK.
Hopp et al., 2021 [42]	30% DCR versus ad libitum	Pkd1^RC/RC^ mice	From 3 to 6 mo of age	Reduced cyst area, reduced kidney weight and fibrosis; Increased p-AMPK/AMPK, not reduced p-S6/S6.
Hopp et al., 2021 [42]	IMF (80% food restriction 3 d/wk, ad libitum other days) versus ad libitum every day	Pkd1^RC/RC^ mice	From 3 to 6 mo of age	No reduction in cyst area; No reduction in kidney weight and fibrosis.
Hopp et al., 2021 [42]	TRF (8 h during the 12 h dark cycle) versus ad libitum	Pkd1^RC/RC^ mice	From 3 to 6 mo of age	No reduction in cyst area; No reduction in kidney weight and fibrosis.
Torres et al., 2019 [41]	TRF (8 h during the 12 h dark cycle) versus ad libitum	Han:SPRD rat	Postnatal weeks 3—8	Reduced cystogenesis and cyst growth; Reduced fibrosis; Reduced activation of mTOR-S6K and STAT3.
Torres et al., 2019 [41]	Ketogenic diet	Han:SPRD rat, juvenile and adult and male versus female	5 wk (age 3–8 wk); (age 8–12 wk)	Reduced kidney weight and cystic indices; Reduced increase in serum creatinine; Reduced activation of mTOR-S6K and STAT3, increased p-AMPK (in the juvenile model); Reduced kidney weight and cystic indices; renal function not affected (in the adult model).
Torres et al., 2019 [41]	Acute fasting	Han:SPRD rat; Pkd1^cond/cond^:Nes^cre^ mice, Feline PKD1 models	48 h fasting with free access to water (Han:SPRD rat); 24 h fasting with free access to water (Pkd1 ^cond/cond^:Nes ^cre^ mice); 72 h fasting with free access to water (Feline PKD1 models)	Reduced cystic area and kidney mass (Han:SPRD rat, Feline PKD1 models); No change in kidney mass (Pkd1^cond/cond^:Nes^cre^ mice).
Torres et al., 2019 [41]	Ad libitum feed + oral BHB	Han:SPRD rat, juvenile and adult and male versus female	5 wk	Reduced kidney mass and cystic area; Inhibition of proliferation; Reduced fibrosis; Improved kidney function.

**Table 2 nutrients-16-02582-t002:** Clinical trials investigating Ketogenic Dietary Interventions in ADPKD patients. DCR, Daily Caloric Restriction; IMF, Intermittent Fasting; KD, Ketogenic Diet; WF, Water Fasting; TRD, Time-Restricted Diet; PFKD, Plant-Focused Ketogenic Diet, TKV, Total Kidney Volume.

Authors, Year of Publication	Intervention	Study Design	Population (N)	Duration	Findings
Hopp et al., 2021 [42]	34% DCR versus IMF (80% restriction every other day)	Pilot clinical trial	Adult overweight or obese ADPKD patients (28)	1 year	Higher adherence with fewer side effects in DCR compared to IMF; Higher weight loss in DCR compared to IMF; Annual change in htTKV qualitatively low versus historical control and correlated with weight loss in both DCR and IMF; No annual change in eGFR in both DCR and IMF; Improved lipid profile only in DCR.
Testa et al., 2019 [47]	Modified Atkins diet	Pilot clinical trial	Adult patients with rapidly progressive ADPKD (3)	3 mo	High rate of overall satisfaction; Increase in total cholesterol levels.
Strubl et al., 2022 [48]	Self-initiated KD and/or TRD	Retrospective case series study of self-reported observation and medical data	Adult ADPKD patients (121)	≥6 mo	Improvement of overall health with both KD or TRD; Amelioration of ADPKD-related symptoms and weight loss more pronounced in KD compared to TRD; Stabilization of eGFR.
Bruen et al., 2022 [49]	PFKD including KetoCitra^®^	Real-life experience clinical trial	Adult ADPKD patients (24)	12 wk	High rate of adherence and satisfaction; 50% of pts refer amelioration of ADPKD-related symptoms; Weight loss; Improved renal function.
Oehm et al., 2023 [50]	3-day WF or a 14-day KD	Prospective interventional short-term clinical trial	Adult patients with rapidly progressive ADPKD (10)	3 days (WF) 14 days (KD)	High rate of satisfaction; No change in TKV; Weight loss; Increased levels of total and LDL cholesterol in KD; Increased uric acid levels.
Cukoski et al., 2023 [51]	KD versus WF (3 of 14 d) versus ad libitum diet	Exploratory, randomized, open, single-center, three arm dietary intervention study	Adult ADPKD patients (63); randomized to KD (23), WF (21), and ad libitum diet (19)	12 wk	Increased eGFR in KD group; Significant weight loss only in KD group; Decrease ht-TKV in KD group (*p* = 0.08); Increased eGFR in KD group (*p* = 0.00); Increased cholesterol and uric acid levels in KD group.

**Table 3 nutrients-16-02582-t003:** Overview of dietary interventions proposed for the treatment and management of ADPKD, their effects on physiopathologic mechanisms, available studies, conclusions, and the status of related clinical trials. ADPKD, Autosomic Dominant Polycystic Disease; mTOR, mechanistic Target of Rapamycin; AMPK, Adenosine Monophosphate-Activated Protein Kinase; STAT3, Signal Transducer and Activator of Transcription 3; BHB, Beta-hydroxybutyrate; CKD, Chronic Kidney Disease, RAAS, Renin–Angiotensin–Aldosterone System; V2R, Vasopressin-2 Receptor; cAMP, cyclic Adenosine Monophosphate; RCT, Randomized Clinical Trial.

Dietary Intervention	Proposed Mechanism	Available Studies	Conclusions	Status of Clinical Trials
Daily caloric restriction	Inhibition of mTOR pathway and activation of AMPK, preventing cell proliferation, fibrosis, and cyst growth.	Preclinical trial [40,41,43] Pilot clinical trial [42]	Difficult long-term adherence or feasibility; Possible recommendation in overweight or obese patients, under medical monitoring.	No published data yet (NCT03342742); Ongoing clinical trial (NCT04907799).
Intermittent fasting	Inhibition of mTOR pathway and activation of AMPK, preventing cell proliferation, fibrosis, and cyst growth.	Preclinical trial [42] Pilot clinical trial [42]	Not enough evidence to specifically recommend in ADPKD patients.	No published data yet (NCT03342742).
Time-restricted feeding	Reduction in mTOR and STAT3 signaling, preventing cell proliferation, fibrosis, and cyst growth.	Preclinical trial [41,42]	Not enough evidence to specifically recommend in ADPKD patients.	Ongoing clinical trials (NCT04534985).
Ketogenic diet	Inhibition of mTOR pathway and activation of AMPK, preventing cell proliferation, fibrosis, and cyst growth.	Preclinical trial [41] Pilot clinical trial [47] Retrospective case series study [48] Real-life clinical trial [49] Prospective interventional short-term clinical trial [50] RCT [51]	Not enough evidence to specifically recommend in ADPKD patients.	No planned clinical trials.
BHB	Inhibition of mTOR pathway and activation of AMPK, preventing cell proliferation, fibrosis, and cyst growth. Anti-inflammatory effects. Improvement of mitochondrial efficiency.	Preclinical trial [41]	Not enough evidence to specifically recommend in ADPKD patients.	Not recruiting clinical trial yet (NCT06100133).
Reduced protein intake	Reduction in hyperfiltration and vasopressin levels.	Preclinical trial [91], Prospective clinical trial [92], Observational cohort study [93,94], Post hoc analysis of the DIPAK observational data [95]	No specific evidence in ADPKD patients, but in accordance with the current management of CKD.	No planned clinical trials.
Reduced salt intake	Decrease in vasopressin secretion, RAAS activation, and blood pressure.	Observational cohort study [93,94], Pilot RCT [104], Qualitative study [105], Post hoc analysis of the HALT-PKD data [107]	No specific evidence in ADPKD patients, but in accordance with the current management of CKD.	No planned clinical trials.
High water intake	Reduction in V2 receptor activation and cAMP production.	Preclinical trial [115,116,117,118] Pilot study [119,120], Pilot RCT [104], Prospective clinical trial [123]	No specific evidence in ADPKD patients. Recommendation to reduce water intake as eGFR declines, in line with current CKD management.	Ongoing clinical trials (ANZCTR12614001216606; NCT02933268; NCT03102632).
Low caffeine intake	Activation of phosphodiesterases, enhanced hydrolysis of cAMP.	Preclinical trial [130], Retrospective, post hoc analysis of the CRISP cohort [131], Prospective analysis of the Suisse PKD cohort [132]	No evidence in animal models or ADPKD patients.	No planned clinical trials.

## Data Availability

All data generated or analysed during this study are included in this published article.

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
