# Peer review of "Potential Add-On Benefits of Dietary Intervention in the Treatment of Autosomal Dominant Polycystic Kidney Disease"

_nutrients, 2024, doi:10.3390/nu16162582_

Round 1
Reviewer 1 Report
Comments and Suggestions for Authors
This is a very good review by Rosati et al, summarizing the state of the art in dietary intervention in the treatment of ADPKD, highlighting ketogenic intervention as a promising therapy. I really like the Review, because they described the rational pathophysiological trials, also preclinical and clinical in the field, emphasized the side effects of the ketone diet and related therapies, and possible dietary counseling in PKD. It's a good review, although some data should be added and minor revisions addressed. If the authors address those comments, this revision should be accepted for publication.
Major comments
As a take-home message from the review, authors should make a paragraph and table/figure summarizing those dietary mechanisms/inversions indicating which ones have clear, medium, low, or no evidence of being potential treatments for ADPKD (including EK), and the evidence and status of potential clinical trials.
Page 1 Line 34. The authors should include the recent classification of ADPKD and similar ADPKD, with the respective related genes. Indicating the need for further studies in other PKD-related diseases.
Page 2 Line 44. For some reason, reference 4 and the related argument are missing. The authors jump from reference 3 to 5.
Page 10 Line 491. References 136 to 140 are missing in the text and the argument is also related. The authors jump from reference 135 to 141.
Minor comments:
Page 1 Line 34. There is a typo: “polycystin 1”, it should be Polycystin 1.
Page 10, line 469. There is a typographical error: “…cystic proliferation and development.[15] Animal studies…” should be “…development [15]. The animal…".
Page 10 line 499. There is a typo: “… follow-up for 2 years. [70] So…”, should be “… … followed for 2 years [70]. Hence …".
Author Response
We thank the Reviewers for their constructive and insightful comments. All were valuable and very helpful for improving our manuscript. We provide here a point-by-point response to all their suggestions. Revised sections are marked in red in the reviewed version of the manuscript.
Response to Reviewer #1
Comments 1: As a take-home message from the review, authors should make a paragraph and table/figure summarizing those dietary mechanisms/inversions indicating which ones have clear, medium, low, or no evidence of being potential treatments for ADPKD (including EK), and the evidence and status of potential clinical trials.
Response 1: Thank you for this suggestion, which we believe has considerably increased the quality of our article. We have now introduced Table 3, which summarizes the available evidence and any scheduled or ongoing clinical trials regarding all dietary interventions applied to patients with ADPKD. We have also added a sentence in the conclusion section that emphasizes how, at present, there is no evidence supporting strong recommendations for any dietary advice or interventions that can significantly and specifically modify the progression of ADPKD. You can find the added sentence on page 14, paragraph 7, lines 766-768.
Comments 2: Page 1 Line 34. The authors should include the recent classification of ADPKD and similar ADPKD, with the respective related genes. Indicating the need for further studies in other PKD-related diseases.
Response 2: Thank you very much for your advice. As suggested, we have included a few sentences regarding the most recently discovered genes implicated in the pathogenesis of ADPKD and ADPKD-like phenotypes. You can find this additional information on page 1, paragraph 1, lines 41-51.
Comments 3: Page 2 Line 44. For some reason, reference 4 and the related argument are missing. The authors jump from reference 3 to 5.
Response 3: Thank you for pointing out this inaccuracy. We have arranged the references so that they follow a progressive sequence.
Comments 4: Page 10 Line 491. References 136 to 140 are missing in the text and the argument is also related. The authors jump from reference 135 to 141.
Response 4: Thank you for pointing out this inaccuracy. We have arranged the references so that they follow a progressive sequence.
Comments 5: Page 1 Line 34. There is a typo: “polycystin 1”, it should be Polycystin 1.
Response 5: As suggested by the reviewer we have modified this typ.
Comments 6: Page 10, line 469. There is a typographical error: “…cystic proliferation and development.[15] Animal studies…” should be “…development [15]. The animal…".
Response 6: As suggested by the reviewer we have modified as required. Now the sentence refers to citation [20].
Comments 7: Page 10 line 499. There is a typo: “… follow-up for 2 years. [70] So…”, should be “… … followed for 2 years [70]. Hence …".
Response 7: As suggested by the reviewer we have modified as required. Now the sentence refers to citation [75].
Reviewer 2 Report
Comments and Suggestions for Authors
It was an interesting read. However, we feel there are several areas for improvement. Please consider revising them.
The entire review paper should be revised to be clearer, with figures and tables. In particular, it was felt that the information, figures and concepts of the literature discussed were difficult to convey to the reader using text alone. As it is a good topic, more means should be taken to make it easier for readers to understand.
The current manuscript is difficult to convey, including the author's arguments, and should be re-structured and submitted again.
Author Response
We thank the Reviewers for their constructive and insightful comments. All were valuable and very helpful for improving our manuscript. We provide here a point-by-point response to all their suggestions. Revised sections are marked in red in the reviewed version of the manuscript.
Response to Reviewer #2
Comments 1: The entire review paper should be revised to be clearer, with figures and tables. In particular, it was felt that the information, figures and concepts of the literature discussed were difficult to convey to the reader using text alone. As it is a good topic, more means should be taken to make it easier for readers to understand.
Response 1: Thank you for your constructive feedback. Based on your suggestions, we have made several changes to enhance the clarity and readability of our manuscript.
We have incorporated a figure that depicts the metabolic changes in ADPKD and those induced by ketosis. These additions aim to facilitate a visual understanding of the complex metabolic reprogramming described in the first section of the paper.
Additionally, we have included three tables. Specifically, Table 1 and Table 2 present evidence from available studies examining the effect of ketogenic dietary intervention in animal models and ADPKD patients, respectively. Table 3 summarizes the proposed mechanisms of action and evidence from all dietary interventions used in the management of ADPKD.
We believe these enhancements significantly improve the manuscript's clarity and readability. We hope the revised version meets the standards of the journal and addresses your concerns effectively.
Comments 2: The current manuscript is difficult to convey, including the author's arguments, and should be re-structured and submitted again.
Response 2: Thank you for your valuable feedback on our manuscript. We appreciate your insights and apologize for any difficulties in conveying our arguments. We understand the importance of clarity and structure in presenting our research effectively. Accordingly, we have undertaken the following modifications:
- We included an illustration and three tables to better convey our data and key points.
- We performed English language revision to enhance readability and clarity.
- We deleted duplications and redundant sentences to streamline the text.
- We reinforced the conclusion section to emphasize the lack of evidence regarding dietary interventions that could slow the progression of ADPKD. We also highlighted the necessity for further studies to provide strong evidence and subsequent treatment recommendations. Furthermore, we provided insights into the comparison of the efficacy of ketogenic dietary interventions and new drugs (e.g., SGLT2 inhibitors and Metformin) and discussed the concurrent use of KDIs and Tolvaptan.
We believe these changes improve the overall quality of the manuscript.
Reviewer 3 Report
Comments and Suggestions for Authors
This is a pertinent and well thought out manuscript about the potential utility of ketogenic or low carb diets as a way to reduce the severity of renal cytogenesis in ADPKD.
Minor editing is needed.
It would be very useful if the authors could discuss the potential of concurrent ketogenic diet and current ADPKD medications (e.g., tolvaptan) for treatment of this disorder.
A graphic abstract depicting the changes and pathways that are altered in ADPKD and the alterations induced by ketogenic diet would be quite helpful.
Comments on the Quality of English LanguageThe manuscript is readable; however, minor English editing to make it easier to read and follow would be helpful.
Author Response
We thank the Reviewers for their constructive and insightful comments. All were valuable and very helpful for improving our manuscript. We provide here a point-by-point response to all their suggestions. Revised sections are marked in red in the reviewed version of the manuscript.
Response to Reviewer #3
Comments 1: It would be very useful if the authors could discuss the potential of concurrent ketogenic diet and current ADPKD medications (e.g., tolvaptan) for treatment of this disorder.
Response 1: Thank you for your insightful comments. We agree with your observations and, as emphasized in the conclusion section of our paper, it is crucial to understand whether combining ketogenic diets with Tolvaptan may slow or even prevent disease progression. Among the cited clinical studies, only two reported patients receiving active treatment with Tolvaptan, though the numbers were quite small. In the study by Strubl et al., 15% of patients were on Tolvaptan, but no significant differences in endpoints were observed besides the data on water consumption. In the study by Bruen et al., 55% of patients were treated with Tolvaptan (out of a total of 20 enrolled patients), but subgroup analyses were not performed. Thus, we currently lack solid evidence of the beneficial effect of combining a ketogenic diet with Tolvaptan. Since drugs activating AMPK seems to show encouraging results in reducing cystogenesis in preclinical models of polycystic disease, we cited the role of Metformin. Accordingly, we have modified page 15, paragraph 7, lines 786-801 as follows:” A comparison of the safety and efficacy of ketogenic nutritional therapy versus pharmacologic intervention that induce the raising of ketone levels, such as SGLT2i (not yet approved in ADPKD patients), could be of high interest.
Metformin, a well-known oral hypoglycemic agent widely used for treating type 2 diabetes, has shown encouraging results in reducing cystogenesis in preclinical models of polycystic disease [154],[155]. Metformin's mechanism of action involves activating AMPK, which leads to decreased cell proliferation and fluid secretion, thereby reducing cyst growth [156],[157]. Since both metformin and ketogenic dietary interventions (KDIs) activate the AMPK pathway, it is plausible they could have a combined effect in reducing cysts. However, published data do not show clear evidence of benefits on changes in ht-TKV or eGFR [158],[159]. Further studies are ongoing (NCT04939935; NCT03764605).
Finally, it would be crucial to understand the additive role of concurrent KDIs and Tolvaptan, the only currently available treatment drug for patients with ADPKD. We currently lack solid evidence of the beneficial effect of combining a ketogenic diet with Tolvaptan. Nevertheless, it may be reasonable to hypothesize that they might have an additive effect, possibly acting on different molecular mechanisms to slow cyst growth.”
Comments 2: A graphic abstract depicting the changes and pathways that are altered in ADPKD and the alterations induced by ketogenic diet would be quite helpful.
Response 2: Thank you for your constructive suggestion. We have, accordingly, created a figure which illustrates the main metabolic alterations involved in the pathogenesis of ADPKD, leading to cysts growth and disease progression. Additionally, we have illustrated the metabolic changes induced by ketosis, which are capable of restoring the correct signaling pathways and proper cellular function, ultimately reducing proliferation and cyst growth.
Response to Comments on the Quality of English Language
Point 1: The manuscript is readable; however, minor English editing to make it easier to read and follow would be helpful.
Response 1: We have made minor English editing and hope that this version is clearer and easier to follow.